# Breaking the spiral of silence: News and social media dynamics on sexual abuse scandal in the Japanese entertainment industry

**Tsukasa Tanihara**[1]*, **Mitsuki Irihara**[2], **Taichi Murayama**[3], **Mitsuo Yoshida**[4], **Fujio Toriumi**[5], **Kunihiro Miyazaki**[5]*

**1** Ritsumeikan University, Kyoto, Japan, **2** New York University, New York, NY, United States of America, **3** YOKOHAMA National University, Yokohama, Japan, **4** University of Tsukuba, Tsukuba, Japan, **5** Indiana University Bloomington, Indiana, USA

* tanihara@fc.ritsumei.ac.jp (TT); kunihirom@acm.org (KM)

**Data Availability Statement:** The data underlying the results presented in the study are available

## Abstract

Highlighting minorities and crime survivors through public discourse is essential for their support and protection. However, advocating for minorities is challenging due to the fear of potential isolation from one's social circles. This reluctance contributes to the societal phenomenon known as the "spiral of silence," significantly impeding efforts to support socially vulnerable individuals. This study centers on a pivotal instance where the silence surrounding sexual abuse in the Japanese entertainment industry was disrupted, in which the late company president had allegedly abused idol trainees of the company for decades. Utilizing extensive data from news media and social media, the study probes the engagement dynamics of public attention to this scandal. Results indicate that users on social media provided earlier and greater coverage for this scandal compared to news media outlets. Furthermore, television demonstrated a significant delay in addressing this issue compared to other news media, such as tabloids, magazines, and online news. Regarding social media engagement, idol fans exhibited a more subdued response to the issue compared to the general public. Notably, fans more loyal to the company tended to be slower to mention the issue, with a higher likelihood of standing in defense of the perpetrators. Moreover, conflicting attitudes were observed within the fan communities, culminating in an observable "echo chamber" phenomenon. This paper presents a novel examination of the process of disruption of social silence and offers critical insights for aiding vulnerable individuals in environments dominated by an unacknowledged spiral of silence. This study is novel in that it suggests a reinterpretation of the "spiral of silence theory" in the age of social media, through a comprehensive analysis of relevant social media data and news media data. This contributes to the body of research that has analyzed the spiral of silence theory online.

## 1 Introduction

In March 2023, the biggest scandal within the Japanese entertainment industry emerged, implicating Johnny Kitagawa, the late former president of Johnny & Associates, Inc., Japan's

from Zenodo, https://doi.org/10.5281/zenodo.10214997.

**Funding:** The author(s) received no specific funding for this work.

**Competing interests:** The authors have declared that no competing interests exist.

foremost male idol management company that dominated the male talent segment of the Japanese entertainment industry with a near-monopoly [1–3]. Posthumous investigations revealed accusations of Kitagawa engaging in sexual abuse toward his idol trainees spanning more than five decades [1]. Reports indicate almost 500 individuals are considered survivors [4]. An independent panel of experts declared this unprecedented crime as "extremely malicious" [5]. After the scandal was revealed in March 2023, the public attention to the incident extended beyond the borders of Japan, with consistent coverage from international media outlets such as the BBC and CNN [6, 7], which ultimately led to investigations and press conferences conducted by the United Nations Human Rights Council [8]. As of November 2023, the company underwent a de facto disbandment, compelling affiliated idol groups to rebrand [9].

A salient aspect of this scandal is the indifference of the news media and public to such serious sexual crimes, which remained largely unacknowledged for over five decades. While accusations of sexual abuse targeting Johnny Kitagawa have previously emerged [10], these claims were often relegated to speculation due to inadequate societal investigation and discourse [11]. This prevailing attitude, allegedly perpetuated by both the news media and the public, fostered an atmosphere of apathy, inhibiting the acknowledgment of these allegations as an urgent societal matter. Notably, both the BBC, instrumental in spotlighting the issue, and the United Nations, during their subsequent investigation, identified the "silence of news media" as a significant obstacle to the adequate acknowledgment and resolution of the issue [8, 12]. This collective public silence aligns with the "spiral of silence" theory (see Theoretical framework), wherein the prevalence of dominant public opinion suppresses the vocalization of opposing perspectives or unpopular narratives [13]. Such silence represents a significant impediment to the advocacy for and protection of socially vulnerable groups, including sexual abuse survivors, and necessitates thorough examination and understanding.

Therefore, we aim to unravel the dynamics that elevated Johnny Kitagawa's alleged sexual abuse issue to become a significant social problem. Our analysis primarily focuses on the role of news media and social media, both central to the "spiral of silence" theory, and especially the behavior of idol fans, a key demographic in this context. Given the potential latent spirals of silence globally, examining instances where such silences are disrupted provides insights into amplifying marginalized voices and supporting overlooked vulnerable groups in societal discussions.

We utilize 14,808 news media articles and 14.5 million social media posts regarding Johnny & Associates, spanning a six-month trajectory from the initial BBC report that incited public discourse to the culmination marked by the expert team's comprehensive report. The analytical framework of this research is threefold. First, by quantifying and visualizing the volume of news media and social media discourse, we trace the chronological emergence and amplification of the issue, aiming to identify the key stages of public awareness and engagement. Second, we classify news media sources, including newspapers and television, to assess which segments acknowledged the issue earlier or later. Third, through the categorization of social media users into diverse groups, idol fans included, we seek to understand the distinct roles these segments played in breaking the prevailing silence.

Our findings indicate that social media users initiated and engaged more extensively in discussions about this issue prior to Japanese news media. Furthermore, television exhibited a significant delay in coverage. Among social media users, it was evident that fans had a more reticent reaction compared to the general populace. Within the fanbase, there was a discernible division in attitudes, giving rise to what can be characterized as an "echo chamber" phenomenon, wherein individuals were exposed predominantly to opinions that mirrored their own. Fans more loyal to the idol groups that currently belong to the company were not only slower to respond but also tended to align with the alleged perpetrators when they did engage.

This study has strengths in observing phenomena using large datasets from news media and social media, as opposed to previous research on the spiral of silence that primarily focused on surveys and interviews with samples. This approach allows for suggesting new developments in the "spiral of silence theory."

## 2 Theoretical framework

The "spiral of silence" theory, first proposed by Noelle-Neumann [13], addresses the phenomenon of media and public silence. The spiral of silence theory is based on the premise that individuals are always conscious of the opinion climate in society. This theory posits that individuals, fearing social isolation, may choose not to voice opinions they perceive as less popular or contrary to the majority view. People fear becoming isolated when they recognize that they belong to the minority, leading them to remain silent. While Noelle-Neumann [13] originally emphasized news media's role in shaping opinion climates, subsequent research [14] expanded this view to include the influence of reference groups, i.e, the populations from which individuals refer to opinion climate, as well. The spiral of silence theory underscores a critical risk inherent in the formation of public opinion: the predominant visibility of majority viewpoints and the consequent suppression of minority perspectives, potentially impeding the cultivation of a robust marketplace of ideas. This phenomenon represents a significant impediment to the advocacy for and protection of socially vulnerable groups, including sexual abuse survivors.

With the media landscape's evolution, particularly the rise of online platforms, recent scholarship has interrogated the applicability of this theory within digital environments. Studies indicate that social media not only act as contemporary public forums for opinion expression but also serve as alternative reference groups for individuals assessing opinion climates [15–20]. These findings suggest the persistent relevance and influence of the spiral of silence in shaping discourse, even amidst the diversification of media channels.

When applying the issue of Johnny Kitagawa, the sexual harassment problem was long *silenced*. Although there were a few who accused him, the topic never gained traction in the news media [10, 11]. This state changed due to some factors in 2023, and it is considered that the silence was broken. In other words, a change occurred in the opinion climate regarding this issue, and speaking out about this problem became the majority. As of late 2023, Japanese news media and social media are consistently abuzz with discussions related to Johnny's issue, indicating a significant shift in public discourse and media attention. Our analysis unravels how the break in the spiral of silence led to this long-ignored issue into a widely discussed topic. Utilizing an extensive dataset from social media platform X and archives of Japanese news articles, we aim to dissect the complexity of this transition, seeking to understand the nuanced dynamics that transformed a long-ignored issue into a subject of the national conversation.

## 3 Literature review

### 3.1 Empirical analysis of the spiral of silence

Extensive research on the spiral of silence theory predominantly centers on its application within political communication [16, 21–25]. These studies typically employ survey designs to empirically validate the theory, using regression analyses to ascertain relationships between individuals' willingness to express support for political candidates, their perception of the prevailing opinion climate, and their level of fear regarding social isolation. For instance, Kushin et al. [16] found the significant role of fear of isolation in the prediction of individuals' expression of support for Hillary Clinton or Donald Trump during the 2016 US presidential election.

Similarly, Chan [21] observed, within the context of Hong Kong's 2015 elections, a negative correlation between fear of isolation and self-censorship, noting that individuals were more likely to express their opinions within groups of like-minded political ideologies. However, certain research indicates the theory's constraints, suggesting its diminished applicability among individuals with pronounced attitude certainty [23, 26].

In addition to survey methods, qualitative approaches such as interviews have also been instrumental in exploring the spiral of silence theory. For instance, Fox and Warber [27] conducted interviews with 52 participants before and after the 2012 U.S. election, uncovering a spiral of silence among participants whose LGBTQ+ identities were concealed within their social networks, thereby subjecting them to the perceived heteronormative majority. Furthermore, experimental designs have yielded insightful data; Masullo et al. [28] orchestrated an online experiment that revealed individuals' emotional responses, particularly negative ones elicited by rude comments, as pivotal in determining their propensity to engage in discourse, overshadowing the prevailing opinion climate. These findings suggest that emotional reactions may significantly influence the dynamics of the spiral of silence, indicating a nuanced interplay of factors that dictate public expression. The relevance and implications of these findings, particularly in non-political contexts such as the entertainment industry scandal discussed herein, warrant further exploration.

A limited corpus of research has ventured into testing the spiral of silence theory using observational methods on large-scale data sets. For example, Morales [17] capitalized on an event during which an automated bot retweeting Venezuelan President Maduro's tweets was deactivated, leading to a notable surge in critical tweets against the president. This study, using the framework of the spiral of silence theory, posited that public opinion can shift when the promotional efforts of an autocratic regime are interrupted. Despite these advancements, an important gap remains in understanding how *silence* is disrupted and public opinion subsequently escalates, particularly in non-political contexts like the entertainment industry. By meticulously documenting this transition through the analysis of extensive social media data and news archives, the present study seeks to capture the dynamism inherent in social issues, breaking the spiral of silence. Furthermore, this research aims to significantly enrich the field of public opinion studies by elucidating the specific actions undertaken by social media users and news media personnel throughout this transformative process.

## 3.2 The impact of social media on the spiral of silence theory

There exists a small body of literature discussing the impact of social media on the Spiral of Silence Theory. For example, Chaudhry [29] analyzed comments attached to articles about race, racism, and ethnicity on Facebook pages. The results indicated that minority users with discriminatory views did not hesitate to express their opinions. This suggests that the effect of the Spiral of Silence Theory may be limited on Facebook. Furthermore, Laor [30] demonstrated through interview research that Facebook groups serve as a receptacle for diverse opinions. What the interviewees highlighted was the value of closed Facebook groups where they could share opinions different from those in the general society. This study suggests the potential for discourse on social media to be picked up by news media. Sohn and Sohn [31] clarified the behavior of the Spiral of Silence through simulation. They found that while the Spiral of Silence phenomenon is less likely to occur when an individual's social network is small, it becomes more probable as the network expands. The authors suggest from these results that the Spiral of Silence phenomenon might occur in the context of mass media with universal access, whereas the formation of opinions might differ in online environments where opinions are locally concentrated.

The theoretical background for such occurrences can be explained by considering the structure of the Spiral of Silence Theory. The background for minorities to remain silent includes the fear of isolation. That is, the premise is recognizing that one is in the minority and fearing it. However, on social media, due to the existence of echo chamber phenomena or communities that share values like Facebook groups, individuals are more exposed to opinions similar to their own. This means that even if one is in the minority in real society, it is possible to avoid the fear of isolation on social media. In this way, opinions aggregated on social media can be picked up by news media and have the potential to shape public opinion.

### 3.3 #MeToo movement

The issue of Johnny Kitagawa can be contextualized within the broader #MeToo movement, which spotlighted sexual abuse allegations globally. This movement bears notable similarities to the case in Japan, echoing the scandal surrounding film producer Harvey Weinstein, who faced allegations of sexual abuse from over 80 individuals across three decades. Additionally, the #MeToo's impact is further underscored by the recognition of the women who spoke up against sexual abuse, called the "Silence Breakers" [32], which infers a more analogous relationship can be found with the current issue.

In Western countries, extensive studies have been presented on the theme of the #MeToo movement and the media. Benedictis et al. [33] examined how #MeToo was handled in British newspapers, revealing that the newspapers played a significant role in enhancing the campaign's visibility. Benedictis et al. [33] pointed out that, although there were some differences among newspapers, many treated #MeToo positively. However, they criticized the focus on the experiences of wealthy white female celebrities and the lack of discussion on solutions to sexual victimization. Noetzel [34] conducted a quantitative content analysis over two years, from one year before to one year after the #MeToo tweets, of news articles published in four U. S. newspapers. As a result, it was shown that not only physical sexual violence but also verbal sexual violence began to be reported, and there was a shift from a posture of pursuing third-party responsibilities to one of pursuing individual perpetrators' responsibilities. However, the authors have warned against individualizing the issue, emphasizing that sexual violence is a societal problem.

In the literature on the spiral of silence theory and #MeToo and the media, there are few studies that dynamically and detailedly describe the process in which the silence state is broken, and public opinion subsequently gains momentum. This study, by using a large amount of social media data and news records to describe that process, records the dynamism in breaking the social silences, making a significant contribution to public opinion research by clarifying how social media users and news media editors reacted in that process. This study particularly excels in comprehensively analyzing the discourse across diverse mass media (TV, newspapers, tabloids, magazines, online news) and social media in a chronological manner. This effort, as examined in the previous section, provides profound insights into whether social media weakens the effect of the Spiral of Silence Theory. Ultimately, it adds insights into an essential discussion in social sciences regarding the visualization of social issues.

Furthermore, it is worth noting that the #MeToo movement has not gained as much traction in Japan compared to international arenas [35, 36]. Cultural factors allegedly create an environment where public discourse is challenging [33]. In such a constrained atmosphere, the instances of the present paper where societal silence is effectively shattered are infrequent. Therefore, our study's exploration of of spiral of silence e in the context of Japan's unique cultural landscape offers critical insights into the dynamics of silence breaking within movements like #MeToo.

### 3.4 Research question

Previous research has highlighted the importance of "fear of isolation," the relationship between social media and the spiral of silence theory, and the spread of the #MeToo movement in the media. However, these studies faced methodological limitations such as the use of sample data, interview, and simulations. In contrast, this paper utilizes a comprehensive and chronological large-scale dataset to elucidate the impact of social media on the spiral of silence theory. Specifically, it provides a detailed description of the dynamics involved when silence is broken and offers a contemporary interpretation of the spiral of silence theory. The research question is as follows:

RQ: In the context of sexual abuse issues that were silenced by mass media, what role has social media played in shaping public opinion?

This study responds to the above research question by analyzing a large-scale dataset and seeks new developments in the application of the spiral of silence theory.

## 4 Data collection

To conduct an analysis of the news media and public response to the scandal involving Johnny & Associates, we collect a comprehensive dataset comprising Japanese news media articles and social media posts. It should be noted that the collection and analysis of these data comply with the terms of use of the respective data providers. Especially, social media data is not displayed in such a way as to reveal personal information.

### 4.1 News media data

The news media data for this study is sourced from Ceek.jp News (https://news.ceek.jp/), a comprehensive aggregator of virtually all Japanese news media articles. The dataset encompasses various details, including the publisher name, headline, and the initial segment of the news articles that are accessible without a paywall. For the purposes of this study, we extract news articles that contain "ジャニーズ" (Johnny's: the most common naming of Johnny & Associates in Japan) either in the headline or the introductory section of the news content. The timeframe for the articles for this study spans six months, from March 1, 2023, to August 31, 2023, encompassing a total of 14,808 articles. These articles were derived from 219 distinct news domains, including tabloids, newspapers, and televisions, providing a broad spectrum of perspectives and reporting styles on the issue. Here, to concentrate solely on Japanese media outlets, we excluded BBC news from the dataset. The rationales for this process are that BBC is exceptional as a silence breaker in the issue; we could not identify any other foreign news (e.g., CNN) written in Japanese reporting on Janney Kitagawa among the top 150 news domains; it is the domestic media that are considered to have substantial commercial ties to the Johnny & Associates [37].

### 4.2 Social media data

For the social media component of this research, we employ data from X (formally Twitter), a platform that boasts significant popularity in Japan, accommodating approximately 67 million active users, nearly half of the country's population [38]. This figure is roughly on par with the number of daily active users in the United States [39], emphasizing X's substantial reach and influence in the Japanese social sphere.

The data collection process is conducted via X's search API. Our search queries are Japanese keywords (translation in parentheses): "ジャニーズ" (Johnny's) OR "ジャニー" (Johnny) OR "ジャ

ニー" (Johnny) OR "#ジャニーズ" (#Johnny's). This approach is designed to encompass a wide range of public discourse pertaining to the Johnny's scandal. The timeframe for data aggregation is from March 1, 2023, to August 31, 2023, aligning with the period of news media collection. During this period, we amass a substantial corpus of approximately 14.5 million posts. Of these, 11.3 million are reposts (constituting 77.5%), while the remaining posts were a mix of quotes, replies, and original posts. The number of users is 1.5 million.

## 5 Retrieval of sexual abuse posts/news

For our analysis, it is necessary to segregate news and posts associated with sexual abuse allegations from the general contents related to Johnny & Associates. For news media, a Japanese author annotates manually whether a certain news of Johnny's is related to sexual abuse. Then, another Japanese author checked a random sample of 100 cases to confirm that Cohen's Kappa score is 0.857, which can be interpreted as almost perfect agreement [40]. The results showed that 3,066 (26.1%) of the news stories were related to sexual abuse.

As for the social media data, we adopt a keyword-based approach since it is impracticable to annotate all the posts manually. To analyze Japanese sentences, we employ the MeCab [41], a Japanese morphological analyzer, and the NEologd, a Japanese dictionary [42]. The initial step involves isolating all posts that contain the word "性加害" (sexual abuse) from the posts related to Johnny's. Subsequently, we extract the top 50 nouns that appear most frequently within these posts. Upon reviewing the top 50 nouns, we pinpoint eight terms that are highly relevant to sexual abuse. Then, we sample 100 posts incorporating each of these eight words to evaluate their validity to posts about sexual abuse. As a result, we judge five words out of eight–"被害者" (victim), "被害" (damage), "告発" (accusation), "性暴力" (sexual violence), and "性加害" (sexual abuse)–to be words related to sexual abuse with 100% accuracy, and we designate these five words as sexual-abuse words and posts containing as sexual-abuse posts. See S1 Table for an exhaustive list of the 50 most frequent nouns, along with the remainder of the keyword candidates and their respective accuracy rates. The current method does not cover all posts that are related to sexual abuse but are without our keywords, and we acknowledge this point as a limitation of the current study. As a result of these processes, we extracted approximately 2.48 million posts–equating to roughly 17.1% of all posts related to "Johnny's"–that specifically pertained to sexual abuse (87.6% of these are reposts). Here, the number of users was 456,645. Note that since the definition of sexual abuse post/news differs between news media and social media, Fig 1 shown later, shows

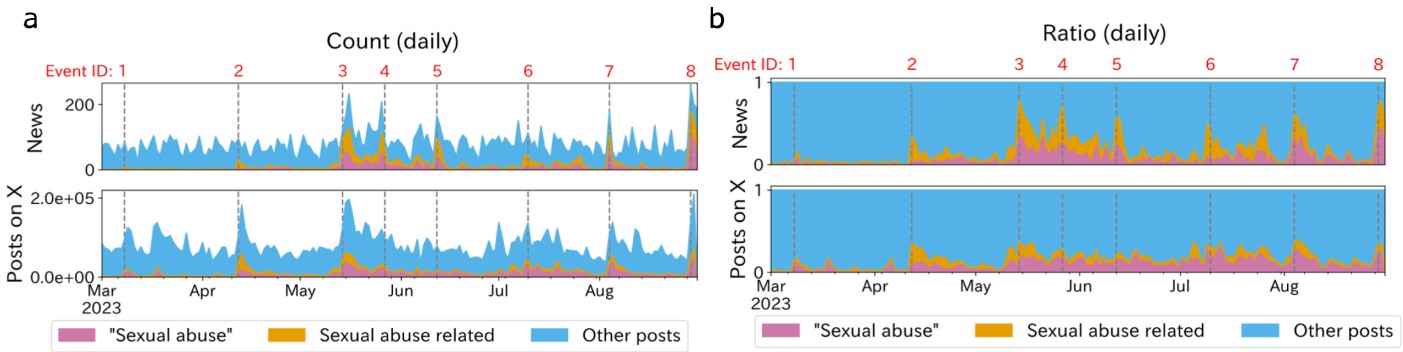

**Fig 1. Volume of news media articles (top) and X (bottom) postings on Johnny's issue. (a)**: Daily volume of Johnny's-related posts/news. **(b)**: Daily counts converted to ratio. In both figures, color emphasizes the posts that contain the word "性加害" ("Sexual abuse": denoted in pink), sexual abuse-related posts but not containing "性加害" (Sexual abuse related: denoted in orange), and other Johnny's-related news (Other posts: denoted in blue). We regarded spikes in the volume that are annotated with grey dashed lines as the main events through their issue. For the details of Event IDs, see main texts.

posts containing the exact word "性加害" (sexual abuse) separately from other sexual abuse-related posts for the purpose of comparison of these two media.

## 6 Trends of reactions to the Johnny's issue in news media and social media

Fig 1 illustrates the temporal variation in the volume of posts on both news media and social media concerning Johnny & Associates. Notable surges in activity, identifiable as spikes in the time series (denoted with dashed lines), correspond to specific events, which are enumerated below:

1. March 8: The British Broadcasting Corporation (BBC) in the UK initiated the wave in Japan with a documentary detailing allegations of sexual abuse committed by Johnny Kitagawa [6]. This pivotal report marked the beginning of widespread attention to the case.

2. April 12: Kauan Okamoto, the survivor, held a press conference at the Foreign Correspondents' Club of Japan, offering a personal account and additional details to the global media, amplifying the issue's visibility [43].

3. May 14: Amidst growing allegations, Julie Fujishima, president of Johnny's company and Johnny Kitagawa's niece, issued a public apology video. This represented the company's first formal acknowledgment of the claims [44].

4. May 27: The Constitutional Democratic Party of Japan, the opposition party in Japanese congress, submitted a proposal for amending the Child Abuse Prevention Law, demonstrating the case's escalating impact on national discourse and policy [45].

5. June 12: A special investigative team dedicated to uncovering the truth and enforcing legal and ethical standards was formed and deployed due to the case's complexity and severity [46].

6. July 10: Tatsuro Yamashita, a famous musician recognized as one of Johnny's most significant business collaborators, publicized his remarks about the sexual abuse allegations, potentially influencing public perception and the entertainment industry's stance [47].

7. August 4: The UN Working Group held a press conference and expressed concern about the hundreds of people involved in the Johnny's sexual abuse and asserted that they are entitled to compensation [48].

8. August 29: The special team presented its fact-findings to the public at a press conference, detailing the contents of the investigative report and possibly influencing subsequent legal, societal, and industry actions, culminating the period of intense investigation and speculation [49].

These pivotal events, reflected as spikes in online engagement and media coverage in Fig 1, signify moments of heightened public interest and media focus, underscoring the case's extensive influence on societal dialogue, legal considerations, and industry practices.

Fig 1 exhibits that the social media users responded more quickly to the scandal than the news media. Specifically, during Event 1 (the BBC report) and Event 2 (the accuser's press conference), there were noticeable spikes in Johnny's-related posts on social media that cannot be seen in news media (Fig 1(a)). Also, in Fig 1(b), social media shows a greater proportion of posts containing the word "性加害" (sexual abuse) than news media: for Event 1, the ratio was 0.10 for social media, compared to 0.01 for news media; for Event 2, the ratio was 0.10 for social media, compared to 0.03 for news media. However, the dynamic shifted by the time of

Johnny's company president's apology video (Event 3), with news media articles with the word "sexual abuse" expanding significantly to reflect 0.30 of all Johnny-related topics, which exceeded social media's 0.15 response ratio (Fig 1(b)). Thereafter (after Event 4), news media continued to largely cover sexual abuse. This analysis underscores a noticeable lag in news media's responsiveness compared to social media, persisting up to the point of direct accusations from the involved parties. Also, we found news media started to cover the issue substantially only after the apology video of the company's president.

## 7 News media reactions by category

The previous section revealed a delayed response to the sexual abuse issue by news media outlets when compared to social media posts. In particular, it has been pointed out that among the news media, television has been particularly silent presumably due to their commercial ties with Johnny & Associates [50, 51], necessitating further examination. Therefore, one of the authors, an expert in communication study, carefully categorizes the 150 most prevalent news domains in the dataset, which collectively constitute 99.8% of all news items in the corpus. These domains were classified into distinct categories (parentheses indicate the ratio of each category among the 150 domains):

- Televisions: News issued by TV companies (6.7%),

- Newspapers: News issued by newspapers focusing primarily on hard news (18.7%),

- Magazines: News issued by magazine publishers (14.0%),

- Tabloids: News issued by newspapers focusing primarily on soft news, such as sports and entertainment (5.3%),

- Online news: News not covered in the categories above, operated by specialized companies for online news (55.3%).

Note that the news by television means a televised story that is republished in the form of online news media.

Fig 2(a) illustrates the counts (top) and ratio (bottom) of news media coverage by media category. The result reveals that online news and magazines initially led the coverage. Television–typically regarded as mainstream media [52]–demonstrated a noticeable delay in their reporting on sexual abuse scandals.

For a closer look, Fig 2(b) plots the number of days elapsed since the BBC reports. The Mann-Whiteney U test results show that television is significantly slower than all four other categories ($p < 0.0001$). Notably, television reports manifested a 36-day gap from the BBC's initial coverage to their first corresponding broadcast. Although TBS [37], one of the largest television station companies in Japan, stated the delay was attributed to the broadcaster's policy of conducting interviews and fact-checking with the perpetrator before reporting, this gap is strikingly larger than other types of media. Looking at the context of #MeToo movement, numerous leading media institutions from left to right, including CNN, Fox news, and the Guardian, initiated coverage of the MeToo movement within a week of its emergence on October 11, 2017 [33, 53], underscoring that silence of 36 days is considerably protracted.

## 8 Social media users reaction by groups

In the previous section, our focus centered on the news media, the primary actor in the spiral of silence. Shifting our lens, we now investigate the corresponding reactions from the other pivotal actor: the public. Thus, we now categorize the X users with sexual abuse-related posts

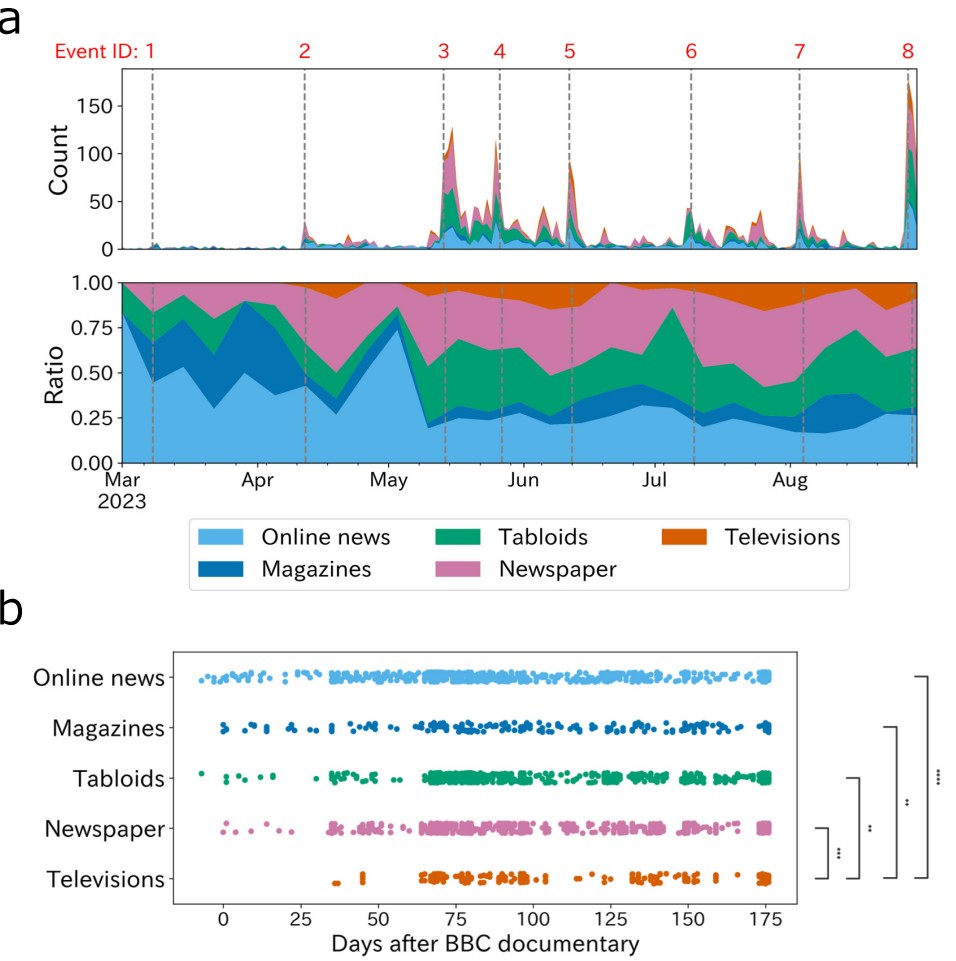

**Fig 2. Transition of news media articles on the issue of sexual abuse by category. (a)**: The top panel shows the counts by category, and the bottom panel shows the ratio. Note that the counts are daily, but the ratios are 7-day totals for visibility. **(b)**: The stripplot the span, in days, of reporting by each category since the March 8 BBC documentary. On the right side are the results of the Mann-Whiteney U test showing the difference between each category. Significance levels are indicated: $p < 0.0001$, 0.001, 0.01, and 0.05 are marked with ****, ***, **, and *, respectively.

and analyze each user group. Due to the sheer number of users and posts in social media data, we use network analysis, topic modeling, and sentiment analysis for these categorizations.

## 8.1 Methods

**8.1.1 User classification by network clustering.** To analyze the user group at the core of the discussion, we create a network of reposts and extract their largest connected components. In this network, the nodes are users who mentioned sexual abuse, and the edges indicate more than a single repost between users, which resulted in the network with 359,109 nodes (78.6% of users who mentioned sexual abuse). Considering that the number of nodes for the second largest connected component encompasses a mere seven nodes, it can be inferred that the rest of the discussions are trivial. To enhance analytical clarity, we eliminate peripheral nodes using k-core decomposition ($k = 3$) [54] and remove the edges with reposts of less than 2 [55, 56]. The resulting number of nodes in this largest connected component is 30,840 of the

users who mentioned sexual abuse. We use Gephi software [57] to visualize the network, and ForceAtlas2 algorithm [58] to shape the network.

We then classify the users using clustering on the created network. Specifically, we adopt the Louvain method [59] for its efficacy in detecting community structure, setting the resolution parameter at one. As a result, we identified 135 clusters and extracted five groups with the largest number of nodes. The five nodes occupy 84.5% of the network (see S2 Table for the ratio of each group).

To understand the characteristics of user groups, we extract the profile text sentences of each user, and after connecting the profile text sentences within a group and considering them as a single sentence, we use the Term Frequency-Inverse Document Frequency (Tf-Idf) method to extract the representative nouns of each group [60]. We manually put the labels for each cluster based on the representative words assigned by Tf-Idf. During this process, we build a stop-word filter to exclude terms that are either ubiquitously Japanese, devoid of substantive meaning, or words directly meaning Johnny & Associates and Johnny Kitagawa, thereby refining the focus on more distinctive, content-rich terms.

**8.1.2 Post classification via topic modeling.** For topic extraction from posts, we employ the biterm topic model (BTM) [61], which is known to work better for short sentences compared to more conventional models, such as Latent Dirichlet Allocation (LDA) [62]. After evaluating perplexity scores, as recommended in Zhao et al. [63], we determine six to be the optimal number of topics for our model. Once we build the topic model, we assign each post to one of the six topics (see S3 & S4 Tables for the labels and representative words of the topics).

**8.1.3 Post classification via sentiment analysis.** We adopt machine learning-based sentiment text classification. While the standard practice often involves classifying sentiments into positive and negative categories [64], the nature of the discourse analyzed in this paper—predominantly negative—necessitated a more nuanced form of sentiment analysis. As such, we employ LUKE-based Japanese emotion analyzer [65, 66] fine-tuned by WRIME dataset [67] that is able to classify Japanese text into one of eight distinct emotional categories: joy, sadness, anticipation, surprise, anger, fear, disgust, and trust. We assign each post to one emotion.

**8.1.4 Comparison of topics and sentiments across user groups.** After the classification of users and posts, we delve into the analysis of the distribution of topics and sentiments across various user groups. However, since we anticipate topics and sentiments to have an inherent distributional bias (i.e., the original volume of posts on some topics and sentiments is originally large), we need to normalize them. Specifically, in the case of topics, we first create a user-topic matrix, where the cells of the matrix are counts, denoted $C_{t, g}$ ($t$ is the topic, and $g$ is the user group). Then, we calculate the expected value $E(t, g)$ for each cell in the matrix, and then calculate the deviation of each sentiment value from that expected value (i.e., residuals) [68, 69]. If this deviation is positive, it means that the sentiment is higher than expected. The expectation and residuals can be calculated as follows:

$$E(t, g) = \sum_t \sum_p C_{t,g} \times \frac{\sum_g C_{t,g}}{\sum_t \sum_g C_{t,g}} \times \frac{\sum_t C_{t,g}}{\sum_t \sum_g C_{t,g}}, \tag{1}$$

$$Residual_{t,g} = (C_{t,g} - E_{t,g}) / \sqrt{E_{t,g}}. \tag{2}$$

In a parallel manner, for sentiments of posts, we calculate the residual from the counts by each sentiment for each group and compare the sentiments between groups.

## 8.2 Results

Fig 3(a) distinctly illustrates the fragmentation of the repost network, mirroring the well-documented echo chamber phenomenon [70], which is represented by groups with pronounced polarities. Such groups typically have opposing viewpoints and tend to communicate predominantly within their own clusters, rarely engaging in cross-group interactions. It is interesting to see the emergence of this phenomenon in the presented case.

Fig 3(b) shows the user profilings derived from the top 10 Tf-Idf words (translated from Japanese; see S5 Table for original terms), revealing a nuanced landscape of fan groups and their affiliations. The groups are composed of there are three types of fans (Fan1 (14.8% of the network nodes), Fan2 (11.6%), and Fan3 (11.3%)) and political users (Politics1 (32.2%) and Politics2 (14.5%)). On social media, political users often join emerging discussions (e.g., [56]), which was true in this case. We see divergent attitudes between fans of Johnny's idols. Fan 1, typified by idol groups like Kinki (https://en.wikipedia.org/wiki/KinKi_Kids) and individuals like Takuya Kimura (https://en.wikipedia.org/wiki/Takuya_Kimura) and Jun Matsumoto (https://en.wikipedia.org/wiki/Jun_Matsumoto), showcases allegiance to the current Johnny &

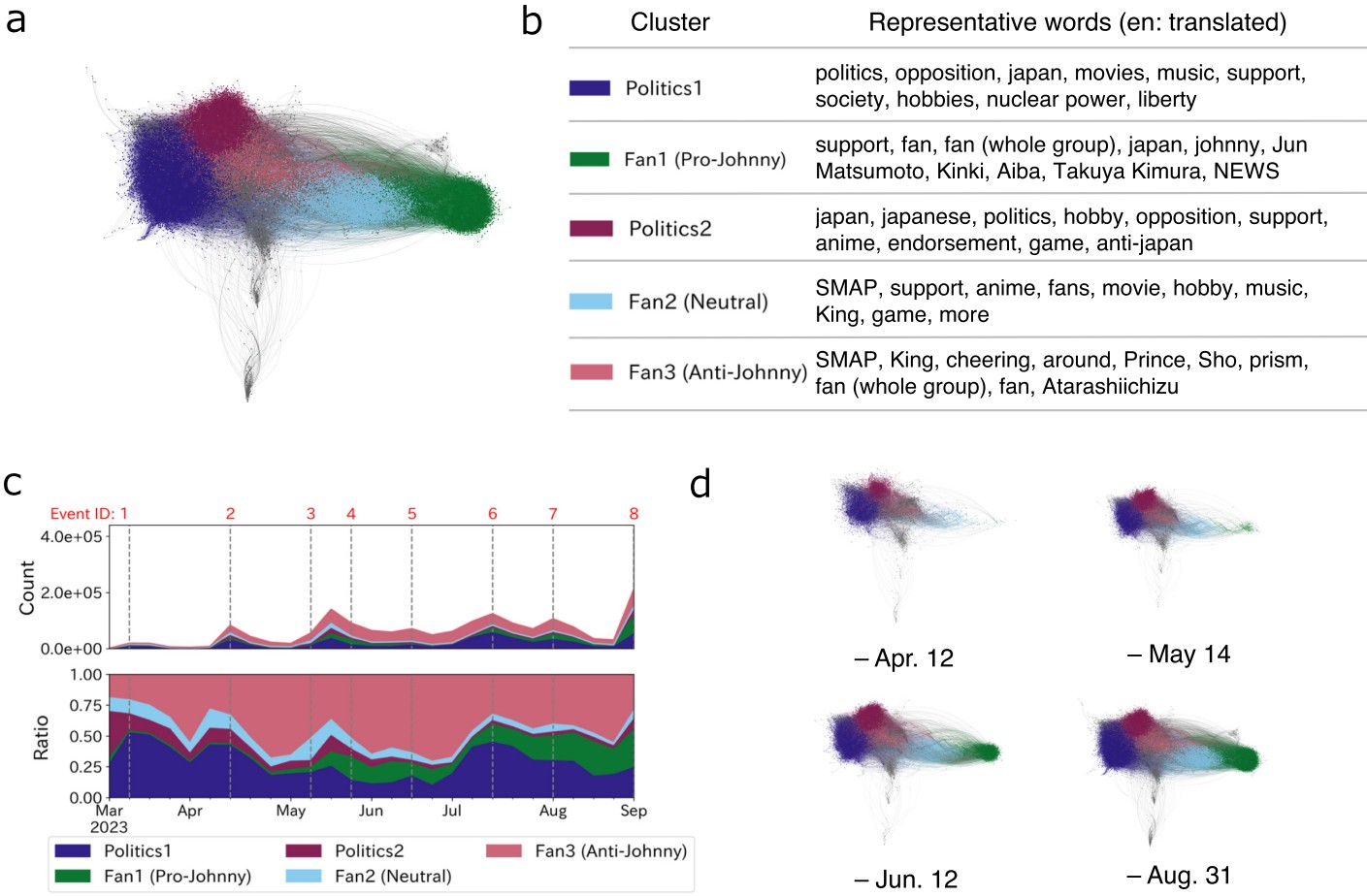

**Fig 3. Network of user groups and the transition of their activity.** (**a**): Network visualization indicating nodes as users and edges as reposts. The color of nodes indicates the cluster that nodes belong to. (**b**): Table depicting the labels of clusters and representative words frequently associated with profiles of groups' users, determined the scores by Tf-Idf. (**c**): Temporal trend of the weekly volume of posts by each group, with the upper chart showing post counts and the lower one presenting the relative ratios. (**d**): Temporal network snapshots showing the repost dynamics on specific dates (Apr. 12, May. 14, Jun. 12, and Aug. 31), indicating increasing polarization over time.

Associates. Conversely, Fan 2 and Fan 3 contain the names of groups such as SMAP (https://en.wikipedia.org/wiki/SMAP) and King and Prince (https://en.wikipedia.org/wiki/King_%26_Prince) that have distanced themselves from Johnny & Associates, either in the past or imminently. This division of the fanbase reflects the situation of Johnny & Associates, where there has been a recent trend of several idol groups leaving the company to pursue independent careers even before the reveal of scandal [71, 72].

By looking at the most shared posts by each group (see S6 & S7 Tables), Fan1 primarily shared the posts claiming "no evidence of sexual abuse" or posts with the hashtag "#ジャニーズ事務所を応援します" (Support Johnny's Office). Notably, even in the face of allegations related to sexual abuse, these posts appear to defend Johnny & Associates and criticize the accusers. On the other hand, the most shared posts by Fan2 are somewhat neutral, many of which simply indicated news that showed the progress of the case. Fan3, contrarily, mainly shared posts criticizing Johnny & Associates and the company's president. Thus, we can tell the stances of Fan1, Fan2, and Fan3 as pro-Johnny's, neutral, and anti-Johnny's, respectively. In the Politics groups, the most shared posts composed of criticisms of varied factors of this issue, including the injustice that only Johnny's company has been allowed for years compared to other scandals, the media and entertainment industry for their silence, the political party's handling of the situation, and feminists. Note that Politics groups are basically anti-Johnny's.

Fig 3(c) shows the temporal volume of posts for each group. It shows that Politics group and Fan2 (Neutral) are the first to respond, followed by Fan3 (Anti-Johnny's). Then, Fan1 (Pro-Johnny's) finally showed up in May, which suggests there is an association between the strength of the relationship with Johnny & Associates and the delay in response.

The chronological breakdown of networks post-BBC documentary paints a compelling narrative (Fig 3(d)). Initially, one side of the polarity dominated the discourse. However, after the apology video of the company's president (May 14) and with the onset of the external investigation (Jun. 12), the other side of the network emerged and eventually shaped the echo chamber phenomenon. This transitionary phase, marked by the looming crisis at Johnny & Associates, potentially urged the current fans into action to protect the company.

In the analysis above, we found a slow response from Fan1 (Pro-Johnny's). Then, how the level of their interest transcend? Fig 4 shows the ratio of sexual abuse posts in each user group in their Johnny's related posts. Fig 4(a) indicates the aggregation over the entire time period

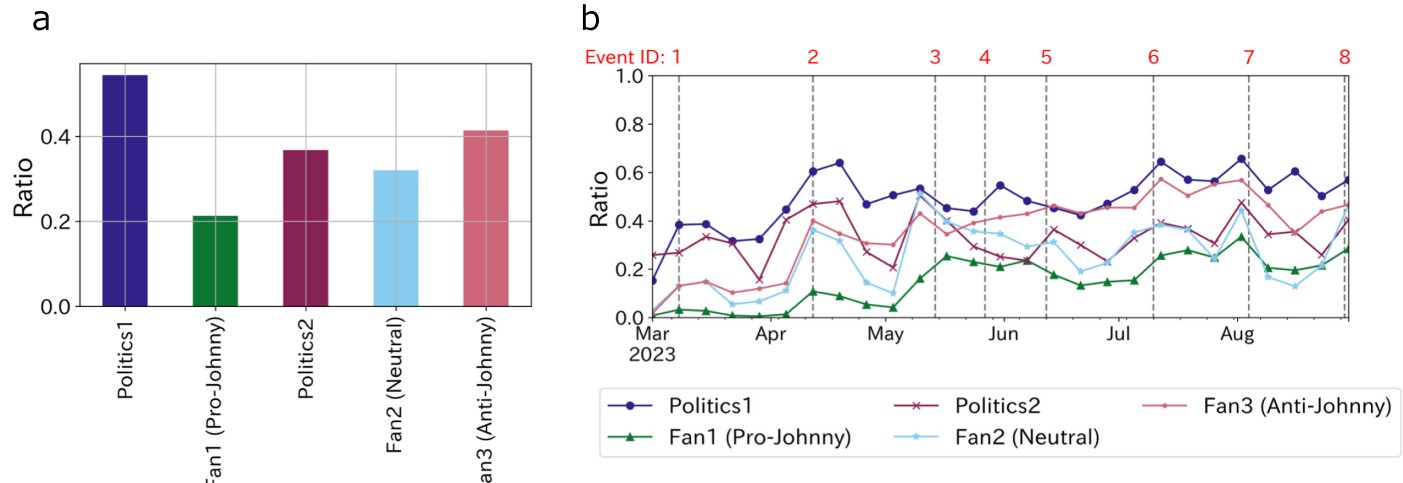

**Fig 4. Ratio of sexual abuse posts in each user group in their posts related to Johnny's. (a)**: The aggregation of the whole period (six months) and **(b)**: transition.

and shows that overall Fan1's engagement with sexual abuse topic is low. Also, Fig 4(b) shows the transition of users' interest in the scandal, which shows that Fan1 (Pro-Johnny's) continued to have low engagement and finally caught up with Fan2 (Neutral) in August, when their engagement was highest. In other words, for a while after the scandal came to light in March, Fan1's interest was not directed toward the scandal, indicating a strong atmosphere of silence regarding this issue.

To gain a deeper understanding of their opinion, we applied topic modeling and sentiment analysis to the posts related to sexual abuse (see Methods for details). For topic modeling, we obtained six topics (parenthesis are the labels of topics that we put):

1. Contents of the press conference of survivors, the investigative team, and the UN group (Press conference)

2. Reports on scandals and investigative findings (News and fact-finding)

3. Company's responsibility and compensation (Responsibility)

4. Johnny Kitagawa and the details of the scandal (Johnny and crime)

5. How the media's surmise exacerbated the scandal (Mass media's surmise)

6. Concerns of fans about the idols (Fans' feelings)

We also perform sentiment analysis to categorize each post into eight sentiments (see Methods). Fig 5 is a heatmap showing how many topics and sentiments each group of posts had. The color of the heatmap indicates the degree of deviation from expectations (see Methods), with pink indicating more posts than expected.

Fig 5(a) reveals that Fan1 demonstrates a marked interest in news and fact-finding, while exhibiting diminished concern for other topics. Contrarily, Fan2 and Fan3 show limited interest in news and fact-finding, with a pronounced focus on corporate responsibility. The Politics category, along with Others, predominantly pertains to the media's surmise and the contents of the crime. The topical interests align with the posts that they most shared, which would lead

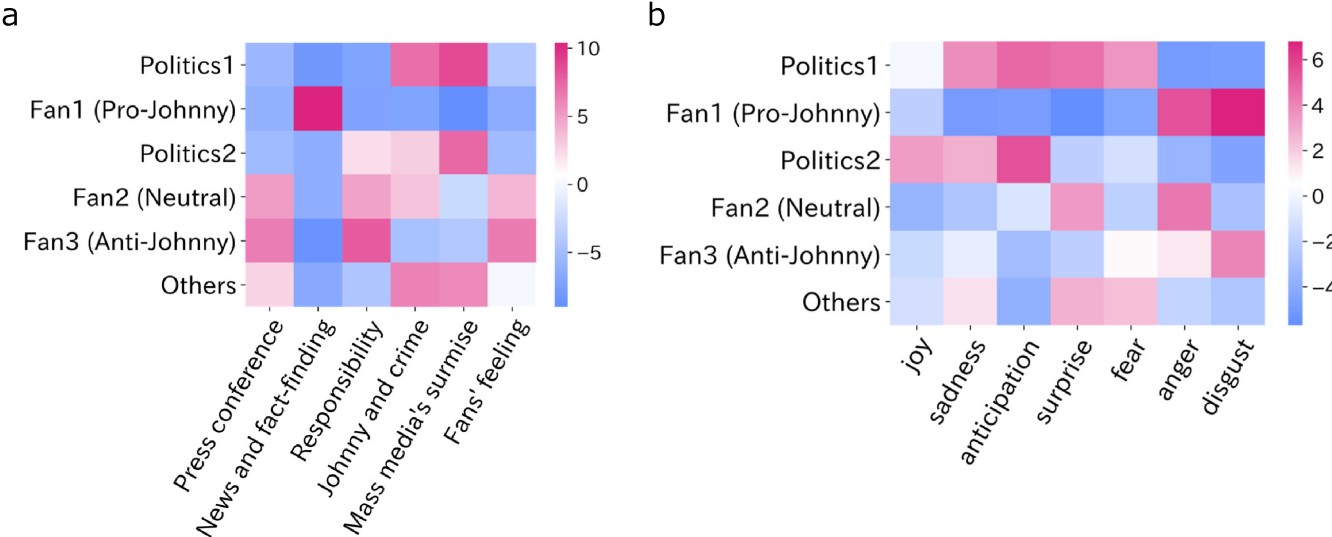

**Fig 5. Heatmap representations of topics-users (a) and sentiments-users (b).** Color of cells indicates the degree of deviation of post counts from expectations in each cell about topics and users (see Residuals). A shift towards pink indicates more posts than expected. Note that we omitted the "trust" from the sentiments due to its exceptionally low counts.

to Fan1's maintained critical perspective on media coverage of the scandals, while Fan2 is interested in the contents of news, and Fan3 frequently critiques Johnny & Associates.

Fig 5(b) indicates the prevalent sentiments among groups. Fan1 predominantly exhibits feelings of anger and disgust, signifying dissatisfaction with news coverage. Fan2 shows heightened surprise and anger, which makes sense considering their interests are directed to the contents of the news. Fan3 expresses disgust, which is seemingly towards Johnny & Associates. Politics groups display heightened anticipation, talking about improving the situation and future.

## 9 Discussion and conclusion

### 9.1 Main results

Compared to traditional news media, users on platform X responded more promptly to Johnny's sexual abuse issue (Fig 1). This suggests that social media users contributed more to breaking the spiral of silence around the matter. This observation aligns with the recent study that social media can undermine the effects of the spiral of silence [29]. Within the social media realm, users often form reference groups, as demonstrated by several studies [15–20]. Specifically, when individuals recognize that their peers on platform X are engaging in discussions of a particular issue, they become more likely to express their viewpoints, regardless of whether mainstream media has acknowledged the issue [18, 73].

Contrarily, mainstream Japanese news media outlets–television–lagged behind the BBC and X users in reacting to the scandal (Fig 2). This study thus offers quantitative evidence of the "silence of news media" highlighted in various news articles and reports [5, 8, 74]. Also, the pace of Japanese media coverage stands in contrast to the rapidity with which Western media outlets reported on the MeToo movement [33, 53]. Several potential factors may explain this discrepancy. For example, in addition to the suspected collusion between the mainstream media and the entertainment industry, Japan may have had few proponents of the debate, such as the feminists who supported the #MeToo discussion in Western countries. One way to speed up news media coverage, for example, would be to deregulate the industry and increase the number of television stations [75].

On the other hand, the news media's reaction was most evident following the apology video by the president of Johnny's company (Fig 2(a) and 2(b)). Subsequent to this press conference, the prevailing public opinion began to perceive Johnny's issue as a genuine concern, elevating it to the status of a "social issue." Actually, TBS, a major Japanese media outlet, admitted in a retrospective analysis that its staff initially regarded Johnny's issue as mere gossipp [37]. The progression of this issue—becoming a public concern primarily after the apology video from the company's president, who was aligned with the perpetrator rather than centering on the survivors' accusation—may suggest potential commercial interests intertwining the news media and entertainment industries. Such dynamics require further research and scrutiny.

As for the users on platform X, we found that whether or not their favorite idol groups belong to the company might have influenced their perception of the sexual abuse scandal. Particularly for Fan1 (Pro-Johnny's), escalating reports of the abuse would likely damage the reputation of their favored idol group, which could be a reason to remain silent on the matter. Such inherent biases of idol enthusiasts may play a role in shaping the opinion climate and fostering the spiral of silence. Yet, since idol fans do not have any incentives to break their silence, disrupting this spiral might necessitate external intervention, such as media exposure. Additionally, the departure of the idol group supported by Fan2 and Fan3 might be substantially attributed to the demise of Johnny Kitagawa. An in-depth evaluation of these factors in dismantling the spiral of silence would be necessary.

As depicted in Fig 3(d), echo chambers emerged among user groups. It's critical to highlight that, amidst the controversy, Fan1 (Pro-Johnny's) maintained a minority viewpoint (14.8% of the network nodes), defending Johnny's company. In an environment where the overwhelming majority criticized Johnny's company, these fans stood out as the sole defenders. In this context, they can be categorized as the "hard core" group, as described by Noelle-Neumann [14, 26] in relation to Johnny's controversy. Actually, Fan1, who is pro-Johnny's, had been silent for some time since the scandal came to light (Fig 4). If they were indeed a hardcore group, even if they were aware of the fear of isolation, a strong attachment to the Johnny company could have fostered a high level of conviction in their stance, making them less resistant to expressing opinions divergent from public sentiment [26]. Alternatively, they might have been insulated from the fear of isolation by the echo chamber effect, being surrounded by peers sharing similar views [16].

## 9.2 Social media and the spiral of silence

Originally, the theory of the spiral of silence was posited in an era predating social media. However, the advent and ubiquity of social media have reshaped the communication landscape, necessitating a reevaluation of the theory's applicability. Three salient points from the Johnny's case illustrate this shift:

- Rapid Social Media Reaction: The immediacy with which social media users responded to the BBC report, juxtaposed against the lag in news media response, highlights the agility of digital platforms in disseminating information.

- Echo Chambers: Even when faced with an overwhelming societal narrative to the contrary, the visibility of like-minded opinions within their echo chamber enabled them to voice their support without apprehension.

- Amplification of Opposition: The public opinion condemning Johnny & Associates, which was in the minority before the issue was made visible, became the majority after being reported by the major media. Subsequently, an opposing public opinion emerged in defense of Johnny & Associates as a counter to the majority, which then expanded its network through echo chambers.

Especially at the time the Spiral of Silence theory was proposed, it highlighted the phenomenon where, fearing isolation, people conformed to the majority opinion, leading to a situation where the majority grew increasingly dominant, the minority increasingly marginalized, and the free market of opinions distorted. However, what this study has implicated is a story in which the minority (initially those criticizing Johnny & Associates, and later, those defending it) gains strength potentially through the power of echo chambers. Thus, in the social media epoch, the spiral of silence theory's core tenets requires contextualization. This study is novel in that it clarifies the role of social media under circumstances where silence prevails, through a comprehensive and chronological analysis of the interaction between multiple mass media and social media regarding this issue.

## 9.3 Implications for the media industry

The media fundamentally serves to illuminate societal injustices and elevate them to the status of public discourse. In this context, the BBC, an international broadcaster, assumed this responsibility. Yet, it was the discussion on the social media platform X that catalyzed widespread awareness in Japan. This digital fervor prompted an apology video by the president of

Johnny's, marking a turning point wherein Japanese mainstream media amplified its coverage, solidifying the matter as a nationally recognized issue.

This study suggests that in the current media environment, where social media is widespread, social media can break the silence even if the mainstream media remains silent, provided there is a trigger. On the other hand, it also revealed that reversing the majority and minority can lead to the emergence of an opposition to the new majority, and this opposition can be amplified.

Japan's television industry is notably oligopolistic, with a starkly fewer number of TV stations than countries like the United States [76]. This is attributed to the permissive nature of broadcasting rights, which have become significant vested interests. Given this structure, Johnny's company is a pivotal player, influencing viewer ratings across nearly all TV stations. This entrenched relationship potentially fosters undue influence. Introducing more TV stations, ensuring their independence from specific entertainment entities [75], and promoting inter-station competition could mitigate such concerns.

## 9.4 Limitation and futurework

In this study, we explored the disruption of silence surrounding the most significant scandal in the annals of Japanese entertainment. We acknowledge several limitations of our current approach. Firstly, while our keyword-centric method for identifying posts about sexual assault prioritizes the recall rate, it may overlook posts that don't explicitly contain our chosen keywords but are still relevant to the topic. Employing machine learning models could address this oversight. Additionally, our categorization of users and posts relies on unsupervised learning rather than theory-driven coding, which presents an area for future refinement.

Regarding future research directions, as delineated in our discussion, several pertinent issues emerge. These include the relationship between commercial ties of media and entertainment agencies and the spiral of silence, the influence of Janie Kitagawa's passing on this spiral, and the external pressures imposed on the Japanese media by foreign entities such as the BBC. While these are crucial subjects, they could not be comprehensively addressed given the constraints of our current dataset. A more in-depth exploration of these topics would provide valuable insights into strategies for breaking societal silences and fostering an environment more conducive to supporting the socially vulnerable.

## Supporting information

**S1 Table. The top 50 frequent nouns that appear in X posts related to Johnny's that include "性加害" (sexual abuse).**
(PDF)

**S2 Table. luster names obtained from network clustering and the percentage of size each cluster occupies in the network.**
(PDF)

**S3 Table. Cluster names of network clustering and their representative words (original) from profiles of users in each group.** Representativeness were calculated by Tf-Idf.
(PDF)

**S4 Table. Result of topic modeling, indicating topic names of post contents and their representative words (English).**
(PDF)

**S5 Table. Result of topic modeling, indicating topic names of post contents and their representative words (original).**
(PDF)

**S6 Table. Clusters and the posts each group shared most (English).**
(PDF)

**S7 Table. Clusters and the posts each group shared most (original).**
(PDF)

## Author Contributions

**Conceptualization:** Tsukasa Tanihara, Kunihiro Miyazaki.

**Data curation:** Mitsuki Irihara, Kunihiro Miyazaki.

**Formal analysis:** Tsukasa Tanihara, Taichi Murayama, Kunihiro Miyazaki.

**Investigation:** Tsukasa Tanihara, Kunihiro Miyazaki.

**Methodology:** Taichi Murayama, Kunihiro Miyazaki.

**Project administration:** Kunihiro Miyazaki.

**Resources:** Mitsuo Yoshida, Fujio Toriumi.

**Software:** Kunihiro Miyazaki.

**Supervision:** Mitsuo Yoshida, Fujio Toriumi.

**Visualization:** Kunihiro Miyazaki.

**Writing – original draft:** Tsukasa Tanihara, Kunihiro Miyazaki.

**Writing – review & editing:** Tsukasa Tanihara, Mitsuki Irihara, Taichi Murayama, Kunihiro Miyazaki.

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
