## [Decision Letter · Decision Letter 0]

7 Feb 2024

PONE-D-23-40334Breaking the Spiral of Silence: News and Social Media Dynamics on Sexual Abuse Scandal in the Japanese Entertainment IndustryPLOS ONE

Dear Dr. Tanihara,

Thank you for submitting your manuscript to PLOS ONE. After careful consideration, we feel that it has merit but does not fully meet PLOS ONE’s publication criteria as it currently stands. Therefore, we invite you to submit a revised version of the manuscript that addresses the points raised during the review process. Please submit your revised manuscript by Mar 23 2024 11:59PM. If you will need more time than this to complete your revisions, please reply to this message or contact the journal office at plosone@plos.org. Please include the following items when submitting your revised manuscript:A rebuttal letter that responds to each point raised by the academic editor and reviewer(s). You should upload this letter as a separate file labeled 'Response to Reviewers'.A marked-up copy of your manuscript that highlights changes made to the original version. You should upload this as a separate file labeled 'Revised Manuscript with Track Changes'.An unmarked version of your revised paper without tracked changes. You should upload this as a separate file labeled 'Manuscript'.If applicable, we recommend that you deposit your laboratory protocols in protocols.io to enhance the reproducibility of your results. Protocols.io assigns your protocol its own identifier (DOI) so that it can be cited independently in the future. For instructions see: https://journals.plos.org/plosone/s/submission-guidelines#loc-laboratory-protocols. Additionally, PLOS ONE offers an option for publishing peer-reviewed Lab Protocol articles, which describe protocols hosted on protocols.io. Read more information on sharing protocols at https://plos.org/protocols?utm_medium=editorial-email&utm_source=authorletters&utm_campaign=protocols.

We look forward to receiving your revised manuscript.

Kind regards,

Hans H. Tung

Academic Editor

PLOS ONE

Journal Requirements:

2. In your Methods section, please include additional information about your dataset and ensure that you have included a statement specifying whether the collection and analysis method complied with the terms and conditions for the source of the data.

**Additional Editor Comments:**

As you can see from the reviewers' reports, while they are both very positive about the manuscript, one of them raises several minor suggestions for it to be publishable. Specifically, they encourage you to be more forthright about your contributions to to the literature and explicit about how your hypotheses are derived and stated. In addition, they also suggest you enrich your literature review section. Please read their reports carefully and revise your manuscript accordingly. 

Reviewers' comments:

Reviewer's Responses to Questions

**Comments to the Author**

1. Is the manuscript technically sound, and do the data support the conclusions?

Reviewer #1: Yes

Reviewer #2: Yes

2. Has the statistical analysis been performed appropriately and rigorously? 

Reviewer #1: Yes

Reviewer #2: Yes

3. Have the authors made all data underlying the findings in their manuscript fully available?

Reviewer #1: Yes

Reviewer #2: Yes

4. Is the manuscript presented in an intelligible fashion and written in standard English?

Reviewer #1: Yes

Reviewer #2: Yes

5. Review Comments to the Author

Reviewer #1: This is a very well written and extremely interesting manuscript. I think it is ground breaking in the way it shows the difference in how traditional news media and social media discuss the same case, but also the difference in types of traditional news media (tabloid and such), and also different groups of social media users (loyal – not loyal). I can understand if some of the other reviewers might argue that the theory section is short, but I think the spiral of silence is sufficient, and that it is used in a very good way to cast light on the topic at hand.

Reviewer #2: I have conducted a thorough examination of the manuscript, which endeavors to explore a pivotal instance wherein the prevailing silence surrounding sexual abuse within the Japanese entertainment industry was disrupted. The study delves into the dynamics of public attention engagement in this context. Below, I present my feedback, and I believe that incorporating the suggested revisions will enhance the manuscript's suitability for publication:

1. The research exhibits innovation and is well articulated. However, I suggest that the authors explicitly highlight the novelty and social implications of the study within the manuscript.

2. While the literature review is comprehensive, I recommend incorporating a section addressing the impact of social media on breaking the spiral of silence. For example, consider including a chapter discussing how social media platforms undermine the spiral of silence by facilitating the expression of diverse voices and opinions: Laor, T. (2023), titled "Breaking the Silence: The Role of Social Media in Fostering Community and Challenging the Spiral of Silence," published in the Online Information Review.

3. The research results are noteworthy and intriguing. However, the manuscript lacks explicitly stated research hypotheses and questions. It is essential to formulate these elements based on the insights gained from the literature review.

4. Integrating research questions and hypotheses derived from the literature review will not only enhance the overall quality of the paper but also contribute to a more enriched discussion.

I trust that these suggestions will contribute to refining the manuscript and ensuring its readiness for publication.

6. PLOS authors have the option to publish the peer review history of their article (what does this mean?). If published, this will include your full peer review and any attached files.

Reviewer #1: **Yes: **Dr. Thomas Wold

Reviewer #2: No

---

## [Author Response · Author response to Decision Letter 0]

5 Mar 2024

Response to Editors and Reviewers

Response to Editors

Response: We rechecked the style requirement. We fixed all headings to be sentence case; unified Figure references to Fig X format.

2. In your Methods section, please include additional information about your dataset and ensure that you have included a statement specifying whether the collection and analysis method complied with the terms and conditions for the source of the data.

Response: Based on this point, we have added the following text to the Data Collection chapter:

It should be noted that the collection and analysis of these data comply with the terms of use of the respective data providers. Especially, social media data is not displayed in such a way as to reveal personal information.

Response: We included captions of each figure in the manuscript.

Response: We added the captions for your Supporting Information at the end of your manuscript. Also, we modified the numbering of tables (added “S” to each table number) in Supplementary materials.

Response: We reviewed the reference list complete and correct. We did not find the papers that have been retracted in the list.

Response to Reviewers

1. The research exhibits innovation and is well articulated. However, I suggest that the authors explicitly highlight the novelty and social implications of the study within the manuscript.

Response: The novelty of this paper is as follows: This paper quantitatively elucidates the relationship between social media and the Spiral of Silence Theory, a topic that has recently been brought into discussion, by comprehensively and chronologically analyzing the discourse across multiple mass media (TV, newspapers, tabloids, magazines, online news) and social media using a large-scale dataset. The results reveal that social media can break the silence even if the mainstream media remains silent, given a trigger. Furthermore, it was found that reversing the majority and minority leads to the emergence of an opposition to the new majority, and this opposition can be amplified.

This has been added to the latter part of the Literature Review and the Discussion sections. Specifically, for the latter, it has been added as the third item in the section on Social Media and the Spiral of Silence within the Discussion section.

Regarding social implications, a chapter titled "Implications for the Media Industry" has been established to discuss this matter. By enriching the description in this part, the social implications have been further emphasized.

2. While the literature review is comprehensive, I recommend incorporating a section addressing the impact of social media on breaking the spiral of silence. For example, consider including a chapter discussing how social media platforms undermine the spiral of silence by facilitating the expression of diverse voices and opinions: Laor, T. (2023), titled "Breaking the Silence: The Role of Social Media in Fostering Community and Challenging the Spiral of Silence," published in the Online Information Review.

Response: In light of your comments, a section titled "The Impact of Social Media on the Spiral of Silence Theory" has been added to the Literature Review, where we discuss based on the literature. In this study, we consider that the mechanism by which social media breaks the Spiral of Silence lies in the reduction of the fear of isolation, brought about by being surrounded by like-minded opinions in the online space. The newly added section discusses this theoretically based on the literature and utilizes it in setting the research question. Additionally, it emphasizes that, unlike previous studies, which were limited by partial data use, interviews, and simulations, this paper is a comprehensive and chronological study using a large-scale dataset.

3. The research results are noteworthy and intriguing. However, the manuscript lacks explicitly stated research hypotheses and questions. It is essential to formulate these elements based on the insights gained from the literature review.

Response: In the final part of the Literature Review, the following research question was articulated:

RQ: In the context of sexual abuse issues that were silenced by mass media, what role has social media played in shaping public opinion?

On the other hand, this paper is not of a hypothesis-testing nature but rather explores and describes big data in an exploratory and comprehensive manner. Therefore, no hypothesis was established.

4. Integrating research questions and hypotheses derived from the literature review will not only enhance the overall quality of the paper but also contribute to a more enriched discussion.

Response: By establishing the following research question, the discussion has been enriched:

RQ: In the context of sexual abuse issues that were silenced by mass media, what role has social media played in shaping public opinion?

The role of social media was not only to break the silence but also to create a complex situation where it reversed the majority and minority, leading to the emergence of an opposition to the new majority and amplifying it. To elucidate this, a third item has been added to the section on Social Media and the Spiral of Silence in the Discussion section.

In addition, we should note that in the annotation indicating statistical tests in Fig. 2, we did not erase the one indicating “no significance” in the figure, so we have erased it. However, this does not affect the overall argument.

---

## [Decision Letter · Decision Letter 1]

2 May 2024

PONE-D-23-40334R1Breaking the Spiral of Silence: News and Social Media Dynamics on Sexual Abuse Scandal in the Japanese Entertainment IndustryPLOS ONE

Dear Dr. Tanihara,

Thank you for submitting your manuscript to PLOS ONE. After careful consideration, we feel that it has merit but does not fully meet PLOS ONE’s publication criteria as it currently stands. Therefore, we invite you to submit a revised version of the manuscript that addresses the points raised during the review process.

We look forward to receiving your revised manuscript.

Kind regards,

Hans H. Tung

Academic Editor

PLOS ONE

Journal Requirements:

Reviewers' comments:

Reviewer's Responses to Questions

**Comments to the Author**

1. If the authors have adequately addressed your comments raised in a previous round of review and you feel that this manuscript is now acceptable for publication, you may indicate that here to bypass the “Comments to the Author” section, enter your conflict of interest statement in the “Confidential to Editor” section, and submit your "Accept" recommendation.

Reviewer #2: (No Response)

2. Is the manuscript technically sound, and do the data support the conclusions?

Reviewer #2: Yes

3. Has the statistical analysis been performed appropriately and rigorously? 

Reviewer #2: Yes

4. Have the authors made all data underlying the findings in their manuscript fully available?

Reviewer #2: Yes

5. Is the manuscript presented in an intelligible fashion and written in standard English?

Reviewer #2: Yes

6. Review Comments to the Author

Reviewer #2: Dear Author,

I have thoroughly reviewed the paper, which centers on a pivotal instance

where the silence surrounding sexual abuse in the Japanese entertainment

industry was disrupted, in which the late company president had allegedly

abused idol trainees of the company for decades. Utilizing extensive data

from news media and social media, the study probes the engagement

dynamics of public attention to this scandal.

The paper explores an interesting and worthy research topic. Below are my

comments:

1. In my opinion, it is important for the authors to augment the abstract

and introduction with the novelty of their research, elucidating the

specific contribution to the existing literature.

3. The literature review is comprehensive and articulately presented.

However, I suggest that the author incorporate discussions pertaining

to the spiral of silence and the influence of social networks. Research

indicates that the anonymity afforded by confession groups on

platforms such as Facebook encourages individuals to share

marginalized issues. Furthermore, it has been observed that social

media platforms not only disrupt the spiral of silence but also prompt

mainstream media to cover issues that originate from social networks.

See: Laor, T. (2023). Breaking the silence: the role of social media in

fostering community and challenging the spiral of silence. Online

Information Review.

4. It would be beneficial for the research questions to be delineated within

a specific chapter and grounded in relevant literature. This approach

would enhance the coherence of the conclusions, facilitating a clear

identification of which hypotheses were supported.

5. The presentation of findings, discussions, and implications is

commendable.

Overall, with the incorporation of these suggested revisions, the manuscript

holds promise for publication.

Best regards,

7. PLOS authors have the option to publish the peer review history of their article (what does this mean?). If published, this will include your full peer review and any attached files.

Reviewer #2: No

---

## [Author Response · Author response to Decision Letter 1]

30 May 2024

Response to Editors

Response: We reviewed the reference list complete and correct. We did not find the papers that have been retracted in the list.

Response to Reviewers

1. In my opinion, it is important for the authors to augment the abstract and introduction with the novelty of their research, elucidating the specific contribution to the existing literature.

Response: 

As you pointed out, we have clearly stated the novelty and contribution of this study to the literature in the Abstract and Introduction. The main novelties of this paper are as follows:

-Instead of using sample data, we conducted observation and analysis using comprehensive data from news media and social media.

-Based on the data analysis, we provided a modern reinterpretation of the spiral of silence theory.

2. The literature review is comprehensive and articulately presented. However, I suggest that the author incorporate discussions pertaining to the spiral of silence and the influence of social networks. Research indicates that the anonymity afforded by confession groups on platforms such as Facebook encourages individuals to share marginalized issues. Furthermore, it has been observed that social media platforms not only disrupt the spiral of silence but also prompt mainstream media to cover issues that originate from social networks. See: Laor, T. (2023). Breaking the silence: the role of social media in

fostering community and challenging the spiral of silence. Online Information Review.

Response: 

In the previous revision, we added a section titled "The impact of social media on the spiral of silence theory" to the literature review and discussed it based on the literature. This study posits that the mechanism by which social media breaks the "spiral of silence" lies in the reduction of the fear of isolation, brought about by being surrounded by like-minded opinions in the online space. The newly added section theoretically discusses this based on the literature and uses it to formulate the research questions.

In response to your latest feedback, we have made further additions. Specifically, we have suggested that social media not only has the potential to break the spiral of silence but also can encourage news media to address issues raised on social media.

3. It would be beneficial for the research questions to be delineated within a specific chapter and grounded in relevant literature. This approach would enhance the coherence of the conclusions, facilitating a clear identification of which hypotheses were supported.

Response: 

We have established a section titled "2.4 Research question" following the Literature Review. In this section, we clearly state the strengths of our research approach and the research questions based on related studies.

On the other hand, this study is an observational and exploratory study using large-scale data. The value lies in the exploratory description of the large-scale data. Therefore, we determined that it would not be appropriate to construct mathematical models and set hypotheses deductively, as this might limit the findings. We appreciate your understanding.

---

## [Editor Report · Decision Letter 2]

11 Jun 2024

Breaking the Spiral of Silence: News and Social Media Dynamics on Sexual Abuse Scandal in the Japanese Entertainment Industry

PONE-D-23-40334R2

Dear Dr. Tanihara:,

We’re pleased to inform you that your manuscript has been judged scientifically suitable for publication and will be formally accepted for publication once it meets all outstanding technical requirements.

Kind regards,

Hans H. Tung

Academic Editor

PLOS ONE
---

## [Editor Report · Acceptance letter]

17 Jun 2024

PONE-D-23-40334R2 

PLOS ONE

Dear Dr. Tanihara, 

I'm pleased to inform you that your manuscript has been deemed suitable for publication in PLOS ONE. Congratulations! Your manuscript is now being handed over to our production team.

Kind regards, 

on behalf of

Dr. Hans H. Tung 

Academic Editor

PLOS ONE